# Effect of Job Stress on Burnout among Nurses Responding to COVID-19: The Mediating Effect of Resilience

**DOI:** 10.3390/ijerph19095409

**Published:** 2022-04-29

**Authors:** Yoon Jung Cha, Kang-Sook Lee, Jeong Hee Cho, Ik Soon Choi, Dahyeon Lee

**Affiliations:** 1Department of Health Promotion, Graduate School of Public Health and Healthcare Management, The Catholic University of Korea, Seoul 06591, Korea; chayj44@gmail.com (Y.J.C.); dada2020@catholic.ac.kr (D.L.); 2Seoul National University Bundang Hospital, Seongnam 13620, Korea; nrcjh72@gmail.com (J.H.C.); ischoi66@snubh.org (I.S.C.); 3Department of Preventive Medicine, College of Medicine, The Catholic University of Korea, Seoul 06591, Korea

**Keywords:** COVID-19, novel infectious disease, burnout, job stress, resilience

## Abstract

Background: This study was conducted to evaluate the relationship of job stress, burnout, and resilience of 271 nurses who worked alternately at a university hospital in South Korea Province and a state-designated inpatient ward for COVID-19 in Korea. Methods: The study sample included nurses who worked at a university hospital in South Korea, during the period between February 2020 and May 2021. The participants (*n* = 271) responded to an online survey between April 2021 and 12 May 2021. The questionnaire included information related to job stress, burn out, and resilience. Results: In phase 1 of regression, job stress had a significant negative effect on resilience of recovery (*β* = −0.397, *p* < 0.001). In phase 2, job stress had a significant positive effect on burnout (*β* = 0.513, *p* < 0.001). In phase 3, resilience had a significant negative effect on burnout (*β* = −0.459, *p* < 0.001). Seventy-five percent of burnout was directly associated with job stress, while 25% of burnout was indirectly associated through mediated effects, through resilience. Conclusions: The promotion of resilience would not only serve as the basis for active coping in situations where burnout and stress are severe, but also serve as a basic driving force for actively overcoming them. Further study to cope with stress and reduce burnout at the organizational level should be conducted.

## 1. Introduction

The occurrence of emerging infectious diseases has been increasing due to frequent overseas exchanges and changes in the environment and ecosystems worldwide [1]. A case in point is the novel coronavirus disease, which has been the most recently occurring infectious disease that was first detected in December 2019 in Wuhan, Hubei Province, China.

The coronavirus disease-19 (COVID-19) has spread worldwide due to its continued proliferation [2]. Initially known only as an infectious respiratory disease with an unknown cause, on 9 January 2020, the World Health Organization (WHO) revealed that the disease is caused by a heretofore undetected RNA virus pathogen belonging to the *Coronaviridae* family. COVID-19 is an infectious respiratory disease caused by SARS-CoV-2. There are various respiratory symptoms of COVID-19 such as fever, cough, dyspnea, and pneumonia. Although there are asymptomatic and mild cases, it can be viewed as a potentially fatal infectious disease. As of May 2021, the accumulated number of confirmed COVID-19 patients was 162,208,643 worldwide. In particular, the accumulated number of confirmed COVID-19 patients in Korea was 132,290, the death toll was 1903, and the number of confirmed patients per day was 597, without a significant decrease [3].

COVID-19 continues to spread rapidly in clusters across various places in the com-munity. This raised the necessity for active responses from medical personnel. Among them, the presence of nursing personnel is required the most in various healthcare units, from screening clinics for COVID-19 response to intensive care units [4].

It has been reported that the occurrence of an emerging infectious disease causes severe stress among nurses [5]. Nurses working with emerging infectious diseases must treat diseases more severely than those of ordinary patients and comply with rules concerning special clothes and infection guidelines instead of wearing general nursing uniforms. The severity and intensity of the work of nurses responding to COVID-19 are higher than those of nurses working in general wards [6]. In general, the number of patients per nurse in the general ward is 10–15, and they typically work for 8 h. For intensive care unit nurses, the number of patients per nurse is two to three critical patients and two to three with mild symptoms, and the length of their working hours is also eight. The number of patients per two nurses in the COVID isolation ward is one to two and two to three patients with mild symptoms, and their work is carried out in shifts of two hours for a total of eight hours.

Job stress occurs when individual needs and organizational goals are imbalanced in the process of communicating within and beyond the organization. Nurses experience psychological burnout due to an increased demand for professional knowledge and skills, role conflicts with other medical staff, poor working environments, conflicts with management and employees of other departments, and various other needs of nurses [7]. Moreover, research has reported that nurses working in the intensive care unit, who must meet the complex needs of patients in critical situations, experience a greater sense of burnout and mental distress than nurses in general wards [8].

Psychological capital refers to utilizing constructive psychological strengths to improve one’s performance in a given environment to embody progressive thinking and attitudes [9]. A previous study defined psychological capital as four dimensions: efficacy, resilience, hope, and optimism. Among them, resilience has begun to be widely studied in the field of positive psychology [10]. Resilience is the ability to recover from and overcome distress caused by problems and obstacles in achieving goals [10]. Moreover, resilience enables the achievement of goals and the improvement of performance based on individuals’ positive psychological states.

In response to the emerging COVID-19 pandemic, the Central Disaster and Safety Countermeasure Headquarters in Korea designated hospitals dedicated to infectious disease prevention and public hospitals, to share roles in isolating and treating infected persons in negative pressure isolation rooms [11]. Moreover, Designated Public Relief Hospitals in Korea diagnose and treat respiratory and non-respiratory patients separately to protect them from infection within the facility [12]. Nurses working in state-designated inpatient treatment rooms provide care, to patients diagnosed with new infectious diseases or suspected patients, in a single-room negative pressure isolation ward, wearing Level-D protective equipment [13].

So far, there have been studies to find the influencing factors of burnout in emergency room nurses, cancer ward nurses and intensive care unit nurses. A previous study about COVID-19 response nurses wrote about the association of the sub-factors of job stress and resilience with burnout, but that study did not examine the mediating effect of resilience [14].

This study examines how factors of job stress relate to burnout among nurses working at Designated Public Relief Hospitals in Korea during the COVID-19 outbreak. Moreover, this study determines the mediating effect of resilience on the relationship between factors of job stress in response to COVID-19 and burnout. This study aims to prove such a correlation. Job stress will have a positive correlation with job burnout and a negative correlation with resilience. Resilience will affect job stress and job burnout. The greater the resilience, the better it is expected to be able to respond to job stress and burnout. This study highlights the importance of enhancing the resilience of medical personnel responding to COVID-19.

## 2. Subjects and Methods

### 2.1. Study Sample and Data Collection

The study sample included nurses who worked at a university hospital in South Korea during the period between February 2020 and May 2021. The sample also included nurses working in state-designated inpatient treatment wards in hospitals located in the southern regions of South Korea where confirmed or suspected COVID-19 patients were hospitalized or treated (COVID-19 isolation ward, COVID-19 isolation intensive care unit, and COVID-19 suspect patient isolation ward). A poster containing a QR code was displayed at the hospital so that those interested in participating in the study could gain access to the online survey, which had been prepared in Google Forms. A total of 300 people worked at the hospital dedicated to COVID-19. Twenty-nine nurses did not participate, so the total number of participants was two hundred and seventy one. The use of an online survey for data collection was rather effective given the ongoing pandemic. This allowed us to contact participants by phone to answer any questions related to the study, which eliminated the need for any face-to-face interaction.

This study was approved by the Institutional Review Board of Seoul National University Bundang Hospital (IRB NO. B-2012/652-305).

### 2.2. Measures

#### 2.2.1. Socio-Demographic Variables

Participants responded to a survey regarding their socio-demographic characteristics such as gender, age, years of service, educational background, and annual salary. Gender was classified into “male” and “female”. Age was classified at 10-year intervals from the “20s” to the “60s”. The number of years of service at the current hospital was categorized as “less than one year”, “less than one to three years”, “less than three to five years”, “less than five to ten years”, and “more than ten years”. Educational background was classified as “high school education or lower”, “associate degree”, “bachelor’s degree”, “master’s degree”, and “doctoral degree”, while the annual salary level was classified at intervals of 16,000 dollars into five groups, from “less than 16,000 dollars” to “more than 65,000 dollars”.

#### 2.2.2. Korean Occupational Stress Scale (KOSS)

A previous study developed a scale to measure and identify factors causing job stress among the Korean population [15]. Considering the number of questions in the entire survey, an abridged version of the scale was used, which excluded factors of job climate and job insecurity and reduced the number of questions constituting each sub-factor. KOSS consists of 30 questions on a 5-point Likert scale ranging from of “strongly disagree”, to “strongly agree”. Items (Cronbach’s α) were included such as physical environment (0.313), on-job demands (0.828), lack of job autonomy (0.205), relationship conflicts (0.757), organizational system (0.878), and inadequate compensation (0.790).

#### 2.2.3. Scale of Burnout

Burnout was measured using a scale consisting of 15 questions, adapted and validated by Shin [16] for Korean workers. This scale is based on the Maslach Burnout Inventory-General Survey (MBI-GS) developed by [17]. Shin’s study secured the validity of the 15-item scale by deleting one question on impersonalization with the lowest internal consistency [16]. Thus, the scale included a total of 15 questions with five questions on emotional exhaustion, four on cynicism (impersonalization), and six on job achievement reduction. Participants responded on a five-point Likert scale ranging from “not at all” to “very much so”. Since the orientation of the items regarding job achievement reduction is described in a positive manner, responses to these items were reverse scored to facilitate interpretation. Cronbach’s α for each item was emotional exhaustion 0.890, cynicism (impersonalization) 0.788, and job achievement reduction 0.817.

#### 2.2.4. Korean-Connor-Davison Resilience Scale (K-CD-RISC)

Resilience was measured using a self-reporting scale measuring one’s stress coping capability. The Connor–Davison Resilience Scale (CD-RISC) was developed by [18] and adapted by [19]. This study used the Korean version of the CD-RISC that consists of a total of twenty-five questions, including nine for toughness, eight for patience, four for optimism, two for control, and two for spirituality. Participants responded on a five-point Likert scale. Cronbach’s α for each item was toughness 0.823, patience 0.872, optimism 0.704, control 0.653, spirituality 0.636.

### 2.3. Data Analysis

The SPSS 25.0 program (IBM Corporation, Armonk, NY, USA) was used to analyze the data and calculate the Cronbach’s α for each scale to confirm their reliability. Descriptive statistical analysis was conducted to determine the levels of the research variables, and a correlation analysis was conducted to identify the correlation between the research variables. To determine the relationship between job stress and burnout, regression analysis was conducted according to the mediation effect verification procedure proposed by [6] and re-verified by bootstrapping using SPSS Process Macro. Statistical significance was determined based on a 5% significance level.

## 3. Results

### 3.1. General Characteristics of the Study Sample

Of the participants, 33 men and 238 women were selected and analyzed as subjects. Of the 271 participants, 226 (87.8%) identified as female, 125 (46.1%) were in their 20s, 70 (25.8%) had served for less than 1 to 3 years, 226 (83.4%) nurses had a bachelor’s degree, and 151 (55.7%) received an annual salary of between 32,000 dollars and less than 48,000 dollars (Table 1).

### 3.2. Correlation Analysis

Pearson’s correlation analysis was conducted to determine the relationship between the variables in this study.

Job stress and burnout demonstrated a statistically significant positive correlation (r = 0.499, *p* < 0.001). Job stress and resilience demonstrated a significant negative correlation (r = −0.420, *p* < 0.001). Burnout and resilience demonstrated a significant negative correlation (r = −0.589, *p* < 0.001).

Furthermore, the absolute value of the correlation coefficient between the measurement variables was less than 0.80, whereby there was no issue of multicollinearity (Table 2).

### 3.3. Verification of the Mediating Effect of Resilience in the Relationship between Job Stress and Burnout

In Step 1, gender, a control variable, was found to have a statistically significant negative effect on resilience (*β* = −0.145, *p* < 0.05). Resilience among females was lower than that of males. Job stress, an independent variable, had a significant negative effect on resilience (*β* = −0.397, *p* < 0.001). The explanatory power of the control variables and job stress for resilience were 21.6% (F = 12.114, *p* < 0.001).

In Step 2, it was found that job stress had a significantly positive effect on burnout, a dependent variable (*β* = 0.513, *p* < 0.001). The explanatory power of the control variables and job stress for burnout was 26.9% (F = 16.166, *p* < 0.001).

Finally, in Step 3, it was found that job stress had a significantly positive effect on burnout (*β* = 0.331, *p* < 0.001). Moreover, resilience had a significant negative effect on burnout (*β* = −0.459, *p* < 0.001). The explanatory power of the control variables, job stress, and resilience for burnout was 43.4% (F = 28.758, *p* < 0.001).

Therefore, job stress directly relates to burnout and indirectly relates to burnout through resilience (Table 3).

Zero is not included between the lower and upper limits of the 95% confidence interval for the estimate of the indirect effects of job stress on burnout mediated through resilience. Thus, it may be stated that the mediating effect of resilience on the relationship between job stress and burnout is statistically significant (Table 4).

### 3.4. Summary of the Results of Hypothesis Testing

The results of the hypothesis test conducted earlier are summarized in Table 5.

## 4. Discussion

This study attempted to examine the relationship between factors of job stress, burnout, and resilience among nurses responding to COVID-19. Further, it verified the mediating effects of resilience on the relationship between job stress and burnout.

Factors of job stress had a significant negative effect on resilience. This is consistent with a previous finding where job stress and resilience were negatively related [20]. Moreover, Baek confirmed that the input of resilience offsets negative effects in the presence of job stress [6]. Therefore, while it is necessary to create optimal job environments and forms of work to reduce job stress for nurses responding to COVID-19, it is also important to improve individual resilience to effectively reduce stress.

Job stress is positively associated with burnout among nurses working against COVID-19. This is consistent with findings from a previous study by [21], who investigated the effects on the burnout of state-designated inpatient treatment ward nurses with a focus on job stress [21]. Oh’s study made use of Parker and DeCotiis’s study job stress measurement tool to determine the level of job stress of intensive care unit nurses. The results showed that their stress level was higher than that of general ward nurses [8,22]. On the basis of this finding, it can be inferred that the job stress and burnout of intensive care unit nurses working against emerging infectious diseases are also higher than those of general ward nurses. Increased job stress also leads to an increase in job burnout. Therefore, it is necessary to constantly monitor the professional environment, rest system for the staff, availability of enough manpower, and the individual state of work overload to reduce the job stress of nurses responding to COVID-19.

The study by [21] presented research results where the job stress of intensive care unit nurses was higher than that of general ward nurses, using a job stress measurement tool developed by [8,22]. This demonstrates that the job stress and burnout of intensive care unit nurses working against emerging infectious diseases are higher than those of nurses working in general wards. Most of the participants in this study are in their 20s, and there is no significant difference from 3 to 5 years of continuous work. In previous studies, superiors have a higher job stress due to the burden of role as middle managers [14]. After more than nine years, job stress is less felt as the opportunity to judge independently increases due to the increase in proficiency and position [23].

According to a previous study by [7] that examined the effect of resilience on burnout with a focus on the experience of clinical nurses, burnout among clinical nurses decreases with higher resilience. This finding supports the results of the current study—resilience has a significant negative effect on burnout [7]. This study highlights the need for individuals to improve resilience on their own and at the organizational level, as well as to become a supporting basis thereof. A nurse’s career, work type management, character content activity, resilience enhancement program, and enjoyment of leisure activities are required. These activities can improve the resilience of nurses [24].

Resilience plays a partially mediating role in the relationship between job stress and burnout among nurses responding to COVID-19. Job stress relates burnout through resilience, indicating that burnout decreases with low job stress and a high level of resilience. This is consistent with the results of a previous study by [25], which states that methods of improving resilience should be sought out to reduce burnout [25]. Accordingly, the promotion of resilience would not only serve as the basis for active coping in situations where burnout and stress are severe, but also serve as a basic driving force for actively overcoming them.

Infection control fatigue of nurses affects job stress. In order to improve this, it is necessary to improve the working environment and provide customized training programs for each career [26]. As the risk and workload from unpredictable emerging infectious diseases and various disastrous situations increase for medical personnel, including those responding to COVID-19, there is an increased need for personnel in the medical sector. A previous study suggested that changing personal and organizational situations might help to prevent job burnout of medical personnel coping with emerging infectious diseases. Personal situations refer to increasing the ability to personally overcome burnout through education, while changes in organizational situations refer to changes in the job environment and organizational members [27].

Sufficient medical personnel should be replaced when they take a break by preparing appropriate standards for the intensity of work and rest areas of medical personnel responding to COVID-19. Spaces for them to rest during breaks must be created in the hospitals. This can be accomplished by acquiring enough extra manpower. These measures pertain to the aspects of providing adequate control and resources in the job demands-resources model to for alleviating psychological factors such as job stress, burnout, tension, and anxiety in the job demand resource model [28].

Hospitals must minimize mental and physical damage to medical personnel who are unfairly treated due to overwork. Instead, hospitals need to improve the treatment of medical personnel before focusing on being patient centered. This will positively reduce the effect of factors associating job stress and job burnout among medical personnel. Nurses’ job stress and burnout were found to be higher as the social support was lower. Applying this, social support is expected to have an important effect on resilience [29]. A study on the development of a coaching program to improve the resilience of new nurses was conducted in 2018 [30].

## 5. Conclusions

Based on this study, a personal-level measure was proposed to improve the resilience among individual medical personnel and reduce job stress. However, support from hospitals and organizations is also important. This is because while there is no limit to internalizing resilience from the perspective of medical staff practicing medicine, resilience is influenced by factors responsible for buffering, such as social support and resources possessed by organizations, superiors, and colleagues [31]. Therefore, a wider range of studies must be conducted on organizational-level countermeasures to alleviate job stress in the future.

## 6. Limitation

Since the survey was conducted only in the southern regions of South Korea, generalizing the findings of this study will be limited. It has limitations as only male and female without other in the gender classification were investigated. Future studies should include a larger and more diverse sample of nurses responding to COVID-19, and further compare the degree of job stress and job burnout among nurses working in general wards. As this is a cross-sectional study, it is difficult to determine a causal relation.

## Figures and Tables

**Table 1 ijerph-19-05409-t001:** General characteristics of the study participants.

Variables	Categories	*n*	%
Gender	Male	33	12.2
	Female	238	87.8
Age(year)	20–29	125	46.1
	30–39	91	33.6
	40–49	43	15.9
	50–59	12	4.4
Duration of service(year)	1<	54	19.9
	1–3	70	25.8
	3–5	59	21.8
	5–10	41	15.1
	≥10	47	17.3
Academic background	High school diploma	3	1.1
	Associate degree	30	11.1
	Bachelor	226	83.4
	Master	12	4.4
Annual income (thousand dollars/year)	16–32	73	26.9
	32–48	151	55.7
	48–65	44	16.2
	≥65	3	1.1

**Table 2 ijerph-19-05409-t002:** Correlation analysis of job stress, burnout and resilience.

	Average	Standard Deviations	Job Stress	Burnout	Resilience
Job stress	3.08	0.46	1		
Burnout	2.91	0.52	0.499 (*p* < 0.001)	1	
Resilience	3.52	0.56	−0.420 (*p* < 0.001)	−0.589 (*p* < 0.001)	1

**Table 3 ijerph-19-05409-t003:** Results of mediated effectiveness verification of resilience.

Model	Dependent Variable	Independent Variable	B	SE	*β*	T	*p*	F(R2)
1	Resilience	Gender (female)	−0.248	0.101	−0.145	−2.465	0.014	12.114 (0.216) *p* < 0.001
		Age	0.034	0.046	0.053	0.736	0.462	
		Duration of service	−0.004	0.033	−0.009	−0.113	0.910	
		Academic background	0.099	0.072	0.077	1.365	0.173	
		Annual income	0.066	0.059	0.080	1.118	0.265	
		Job stress	−0.486	0.072	−0.397	−6.776	<0.001	
2	Burnout	Gender (female)	0.062	0.090	0.039	0.692	0.489	16.166 (0.269) *p* < 0.001
		Age	−0.027	0.041	−0.045	−0.647	0.518	
		Duration of service	−0.021	0.030	−0.057	−0.716	0.475	
		Academic background	−0.028	0.065	−0.024	−0.440	0.660	
		Annual income	−0.045	0.052	−0.060	−0.865	0.388	
		Job stress	0.579	0.064	0.513	9.063	<0.001	
3	Burnout	Gender (female)	−0.043	0.080	−0.027	−0.536	0.592	28.758 (0.434) *p* < 0.001
		Age	−0.012	0.036	−0.021	−0.338	0.736	
		Duration of service	−0.023	0.026	−0.061	−0.873	0.384	
		Academic background	0.013	0.057	0.011	0.235	0.814	
		Annual income	−0.017	0.046	−0.023	−0.378	0.706	
		Job stress	0.373	0.061	0.331	6.119	<0.001	
		Resilience	−0.423	0.048	−0.459	−8.749	<0.001	

**Table 4 ijerph-19-05409-t004:** Validate the mediated effectiveness of resilience with bootstrapping.

Path	B	SE	95% CI	
			LLCI	ULCI
Job stress → Resilience → burnout	0.205	0.047	0.126	0.305

**Table 5 ijerph-19-05409-t005:** Summary of hypothesis validation results.

	Hypothesis	Result
1.1	*Job stress will have a significant negative correlation to resilience.*	Adoption
1.2	*Job stress will have a significant positive correlation with burnout.*	Adoption
1.3	*Resilience will have a significant negative correlation to burnout.*	Adoption
2.1	*Resilience will play a role in the relationship between job stress and burnout.*	Adoption

## Data Availability

This study is a self-reported survey, and information was obtained from the subject oneself.

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
