# Peer review of "Effect of Job Stress on Burnout among Nurses Responding to COVID-19: The Mediating Effect of Resilience"

_ijerph, 2022, doi:10.3390/ijerph19095409_

Round 1
Reviewer 1 Report
No other comments.
Author Response
Thank you for your kind recommendation.
We modified the English expression.
Take good care of me.

Reviewer 2 Report
Introduction:
The introduction appears to end with a hypothesis but the hypothesis is not clearly and theoretically developed. Can the authors please rewrite this section to ensure a clear development of the hypotheses including the mediated effects expected and why.
Results:
Using the three step regression approach to test mediation is quite dated. I also feel the way the analysis was conducted could be more thoroughly explained. Using SEM might be a better approach for testing the hypotheses.
Discussion:
Could you comment in the discussion on your finding about resilience, specifically, why do you think it may be lower among your female participants than your male participants, and what are your thoughts on the implications of this finding, in terms of workforce planning/ training/personal and organizational strategies?
How might nurses ‘improve resilience on their own’ (line 237)?
It is interesting that nurses’ age and duration of service has no significant bearing on resilience (noting that almost half of your participants are aged < 29 years). You may think that older nurses with experience built up over the years may be more resilient.
Concern about contracting COVID and/or bringing it home to family may also increase job stress for nurses, especially earlier in the pandemic when less was known about it. This may be worthy of comment.
Are there any studies that have investigated the application of specific (personal and organisational) strategies to develop resilience in nurses, and whether these have been effective?
Authors please consider the following grammatical corrections:
- Line 14:
- ‘…in Gyeonggi-do Province and a state-designated inpatient ward for COVID-19 in Korea.’
- Lines 14-16: The information described as here as Methods is really ‘Results’. For the Methods, please describe the study in a sentence or two (ie. date range of study, type of data collection (online survey), summary of measures/themes, etc.).
- Line 21: The conclusion would benefit from specific detail about your suggestions for easing burnout in nurses through increasing resilience.
- Line 23: Please change to ‘reduce burnout at the organizational level’
- Line 37: ‘Although there are asymptomatic and mild cases, it can be viewed as a potentially fatal infectious disease’
- Line 43/44: ‘This had raised the necessity of active responses from medical personnel. and a large number of medical person.’
- Line 48: ‘Nurses working toward emerging infectious diseases, as they must treat diseases more severely than those of ordinary patients…’
- Line 51: The second paragraph of the discussion (starting ‘The severity and intensity of the work of nurses responding to COVID-19 …’) would be better added here to help the readers understand the specific stressors for nurses working in COVID-19 control. There may be more stressors that you could add here.
- Sentence starting ‘Job stress’ should be a new paragraph.
- Line 67: ‘To prevent damage caused by COVID-19 to the people, In response to the emerging COVID-19 pandemic, the Central Disaster and Safety Countermeasure Headquarters in Korea designated hospitals dedicated to infectious disease prevention, and public hospitals, to share roles in isolating and treating infected persons in negative pressure isolation rooms.’
- Line 77: ‘A Pprevious study…’
- Line 81: ‘…relate to burnout’
- Line 92-94: Please delete the following sentence as the breakdown by male and female should go into the ‘Results’ section, and you have previously mentioned the study timeline.
- The participants (n = 271) responded to an online survey between April 2021 and May 12, 2021. Of the participants, 33 men and 238 93 women were selected and analyzed as subjects.
- Line 98: ‘A total of 300 people worked at the hospital dedicated to COVID-19 where they worked. Of them these, 29 did not participate, so the total number of participants completing the questionnaire was 271.’ with complete the questionnaires.
- Line 99: Please delete this sentence Their responses were subjected to data analysis.
- Line 100: ‘This allowed you researchers to contact them participants by phone if you had to answer any questions related to the study’
- Line 108: Please delete this sentence - A frequency analysis was performed.
- Line 157: Please change from ‘were women’ to ‘identified as female’.
- I think you could expand on your Results section a little. For example, where appropriate, you could name the variables, instead of calling them ‘the variables’.
- Line 161: (Table heading). It would be preferable to say ‘participants’ rather than ‘subjects’
- Line 171: Table 2. ‘Correlation analysis of Jjob stress, burnout and resilience.’
- Line 198: Table 5 describes your summary of hypotheses. It would be helpful to state your hypotheses early in the paper (say, at the end of the Introduction) so that you provide a good framework for what follows.
- Lines 203-209: (Discussion, second paragraph) As mentioned above, this would be better in the Introduction to help describe the stresses on COVID nurses.
- Line 237: I wasn’t sure what was meant by the following …’as well as to become a supporting basis thereof.’
Author Response
Thank you for your kind recommendation.
According to your opinion, we added several things.
Introduction:
The introduction appears to end with a hypothesis but the hypothesis is not clearly and theoretically developed. Can the authors please rewrite this section to ensure a clear development of the hypotheses including the mediated effects expected and why.
: We have revised as follows.
This study aims to prove such a correlation. Job stress will have a positive correlation with job burnout and a negative correlation with resilience. Resilience will affect job stress and job burnout. The greater the resilience, the better it is expected to be able to respond to job stress and burnout. This study highlights the importance of enhancing the resilience of medical personnel responding to COVID-19.
Results:
Using the three step regression approach to test mediation is quite dated. I also feel the way the analysis was conducted could be more thoroughly explained. Using SEM might be a better approach for testing the hypotheses.
: This original article is a dissertation thesis, and it is difficult to analyze the contents again because the narrative has already been completed. I hope you understand. Thank you.
Discussion:
Could you comment in the discussion on your finding about resilience, specifically, why do you think it may be lower among your female participants than your male participants, and what are your thoughts on the implications of this finding, in terms of workforce planning/ training/personal and organizational strategies?
In a previous study, among medical staff in the COVID-19 situation, women had higher levels of anxiety and depression than men. It seems to be related to this phenomenon.(Batra, 2020)
Batra K, Singh TP, Sharma M, Batra R, Schvaneveldt N. Investigating the Psychological Impact of COVID-19 among HealthcareWorkers: A Meta-Analysis. International Journal of Environmental Research and Public Health 2020, 17, 9096
How might nurses ‘improve resilience on their own’ (line 237)?
Line 246: A nurse's career, work type management, character content activity, resilience enhancement program, and enjoyment of leisure activities are required. These activities can improve the resilience of nurses. [25]
It is interesting that nurses’ age and duration of service has no significant bearing on resilience (noting that almost half of your participants are aged < 29 years). You may think that older nurses with experience built up over the years may be more resilient.
Line 236: Most of the participants in this study are in their 20s, and there is no significant difference under 3 to 5 years of continuous work. In previous studies, superiors have a higher job stress due to the burden of role as middle managers [13]. After more than nine years, job stress is less felt as the opportunity to judge independently increases due to the increase in proficiency and position.[24]
Concern about contracting COVID and/or bringing it home to family may also increase job stress for nurses, especially earlier in the pandemic when less was known about it. This may be worthy of comment.
Line 257: Infection control fatigue of nurses affects job stress. In order to improve this, it is necessary to improve the working environment and provide customized training programs for each career. [27]
Are there any studies that have investigated the application of specific (personal and organisational) strategies to develop resilience in nurses, and whether these have been effective?
Line: Nurses' job stress and burnout were found to be higher as the social support was lower. Applying this, social support is expected to have an important effect on resilience. [30] A study on the development of a coaching program to improve the resilience of new nurses was conducted in 2018. [31]
Line 14: in Gyeonggi-do Province and a state-designated inpatient ward for COVID-19 in Korea.
Lines 14-16: The study sample included nurses who worked at a university hospital, in Gyeong-gi-do, during the period of February 2020 and May 2021. The participants (n = 271) responded to an online survey between April 2021 and May 12, 2021. The questionnaire included information related to job stress, burn out, and resilience.
Line 21: The promotion of resilience will not only serve as the basis for active coping in situations where burnout and stress are severe, but also serve as a basic driving force for actively overcoming them.
Line 23: Further study to cope with stress and reduce burnout at the organizational level should be conducted.
Line 37->39: Although there are asymptomatic and mild cases, it can be viewed as a potentially fatal infectious disease.
Line 43/44->45: This had raised the necessity of active responses from medical personnel.
Line 48->49: Nurses working toward emerging infectious diseases, must treat diseases more severely…
Line 51->52: The severity and intensity of the work of nurses responding to COVID-19 are higher than those of nurses working in general wards [18]. In general, the number of patients per nurse in the general ward is 10–15, and they typically work for 8 hours. For intensive care unit nurses, the number of patients per nurse is 2–3 critical patients and 2–3 with mild symptoms, and the length of their working hours is also 8. The number of patients per two nurses in the COVID isolation ward is 1–2 and 2–3 patients with mild symptoms, and their work is carried out in shifts of two hours for a total of 8 hours.
Line 67->75: In response to the emerging COVID-19 pandemic, the Central Disaster and Safety Countermeasure Headquarters in Korea designated hospitals dedicated to infectious disease prevention, and public hospitals, to share roles in isolating and treating infected persons in negative pressure isolation rooms
Line 77->85: A previous study (Shin, 2021)
Line 81->89: This study examines how factors of job stress relate to burnout…
Line 92-94->105-107: Moved the sentence to the Results section and removed the following sentence.
Line 98->108: A total of 300 people worked at the hospital dedicated to COVID-19. Of these, 29 did not participate, so the total number of participants was 271.
Line 99: Deleted in response to feedback.
Line 100->111: This allowed to contact by phone to answer any questions related to the study,
Line 108: Deleted in response to feedback.
Line 157->168: Of the 271 participants, 226 (87.8%) identified as female,
Line 161->172: Table 1. General characteristics of the study participants.
Line 171->182: Table 2. Correlation analysis of job stress, burnout and resilience.
Line 198->92: This study aims to prove such a correlation. Job stress will have a positive correlation with job burnout and a negative correlation with resilience. Resilience will affect job stress and job burnout.
Lines 203-209: Following the feedback, I moved the paragraph to the introduction.
Lines 237-245: as well as to become a supporting basis thereof.

Reviewer 3 Report
The authors have explored the mediating effect of resilience in the relationship between job stress and burnout among nurses responding to COVID-19.
The study in pioneer in the field. The article shows proper rationale. The methodology is adequate. Sample size is acceptable. However, I have a series of comments that might help improve the quality of the study:
Abstract
Revise the following sentences:
“This can be said that..”
“Conclusions: It was suggested that the ways to ease burnout from 21 the job stress of nurses responding to COVID−19 in the process of performing their duties”. It seems that this sentence is incomplete.
Introduction
At the end of the introduction, introduce justification for the study and also, state the hypothesis of the study (referred to in Table 5).
Tables
In Table 5, it would be useful to include means and standard deviations.
Discussion
Terminology used in the last part of the discussion should be revised. The study is cross-sectional; therefore, terms like “must” (lines 256, 266) should be rather replaced for others more consistent with a cross-sectional study.
References
References in the text must be adapted to the format of IJERPH.
Examples: Luthans and Youssef (2007), (Shin, 2021), Jang et al. (2005), Shin (2003), Connor et al. (2003), Baek (2010), Baron and Kenny (1986), Baek (2016), Parker and DeCotiis (1983), Lim (2018), Oh (2016), Kang (2016), etc.
There are several references that appear in the text but not in the reference list: Baron and Kenny (1986), Connor et al. (2003). Authors should check that all the references are cited in the reference list.
Author Response
Thank you for your encouraging remarks and insightful comments on our manuscript. We have tried to respond to your comments. Please find our point-by-point responses below.
Abstract
Revise the following sentences:
“This can be said that..”
“Conclusions: It was suggested that the ways to ease burnout from 21 the job stress of nurses responding to COVID−19 in the process of performing their duties”. It seems that this sentence is incomplete.
: We have revised as follows.
Line 21: 75% of burnout directly associates job stress, while indirectly associating 25% burnout through mediated effects through resilience. Conclusions: The promotion of resilience wouldn’t only be served as the basis for active coping in situations where burnout and stress are severe, but also be served as a basic driving force for actively overcoming them.
Introduction
At the end of the introduction, introduce justification for the study and also, state the hypothesis of the study (referred to in Table 5).
: We have revised as follows.
Line 92: This study aims to prove such a correlation. Job stress will have a positive correlation with job burnout and a negative correlation with resilience. Resilience will affect job stress and job burnout. The greater the resilience, the better it is expected to be able to respond to job stress and burnout. This study highlights the importance of enhancing the resilience of medical personnel responding to COVID-19.
Tables
In Table 5, it would be useful to include means and standard deviations.
: The above information is shown in Table 2.
Discussion
Terminology used in the last part of the discussion should be revised. The study is cross-sectional; therefore, terms like “must” (lines 256, 266) should be rather replaced for others more consistent with a cross-sectional study.
: We have revised as follows.
Line 267: Sufficient medical personnel should be replaced when they take a break by preparing appropriate standards for the intensity of work and rest areas of medical personnel responding to COVID-19.
We have revised the format of the reference to suit the regulations. Thanks for the feedback.

This manuscript is a resubmission of an earlier submission. The following is a list of the peer review reports and author responses from that submission.
Round 1
Reviewer 1 Report
The article is interesting, and the subject of interest to the reader. But I have a concern related to the method used. In terms of the measure scales used, the authors based their study on validated measurement tools (KOSS, Scale of Burnout, K-CD-RISC). But the results obtained were analyzed as 3 single constructs. For example, for KOSS, the scale contained 30 items, containing several sub-factors. The authors then present the use of this scale in a single construct with a Cronbach's alpha of 0.892. But we cannot use this scale in a single construct without checking its unidimensionality. The correct method would be to perform a factor analysis to verify the unidimensionality of the construct. In the case where this is not to demonstrate, the scale cannot be used as a single variable, a single construct. The same was done for the Scale of Burnout and the K-CD-RISC. This gap influences the validity of all the results of the study.
Reviewer 2 Report
The present study examined how occupational stress factors affect burnout among nurses working in designated public hospitals in Korea during the COVID-19 outbreak. In addition, the authors determined the mediating effect of resilience on the relationship between occupational stress factors in response to COVID-19 and occupational burnout. Participants (n = 271) responded to an online survey. The paper covers a very interesting and contemporarily important area of research related to COVID-19. According to the authors' findings, occupational stress affects occupational burnout through resilience, indicating that occupational burnout decreases with low levels of occupational stress and high levels of resilience.
This study also highlights the need for individuals to improve resilience, both at their own level and at the level of the organisation and also to create a supportive base for those working at COVID-19.
To complement and enrich the work presented, I offer some comments for consideration by the authors:
1/ The authors write that 33 men and 238 women were selected from the participants and analysed as study subjects. The authors do not provide a criterion for selection. Did more respondents complete the questionnaires? How many respondents were rejected and under what conditions? This information is worth including in the manuscript.
2/ In my opinion, it is worth including a "limitation" section and describing the limitations of the study more elaborately. Could the form of filling in the questionnaires "via the internet" have influenced the study? Were the questionnaires used designed to be used in the web version or just paper-and-pencil? were they verified in the web version? might it be worth checking Cronbach's ALFA minimum for the results?
Reviewer 3 Report
This manuscript is believed to be a study demonstrating the importance of exhaustion in nurses caring for patients with COVID-19. However, there is no clear difference between nurses with COVID-19 infectious disease compared to previous studies examining nurse burnout, job stress, and resilience.
Round 2
Reviewer 1 Report
Thank you for revisions and including Cronbach's apha data for sub-factors (note that vocabulary should be unified, as a "sub-factor" and an "item" could not be used indiferently - a sub-factor is constituted of items - ).
But even if Cronbach's alpla are presented for sub-factors, this does not allow to analyse each variable as a whole. Ex: for the Scale of Burnout, there are 3 sub-factors (emotional exhaustion, cynicism and job achievement reduction). Those are constructs of items that should be analysis separately. You can not added them in a single construct named as Burnout, like it as been done in Tables 2 and 3. It is the same of KOSS and K-CD-RISC. For KOSS, initial study (Jang, 2005) created a tool of 43 items from which they extracted 8 sub-factors. You used 6 of them (30 items) for the present study. You have to analyse the results in light of 6 variables, and not of only 1 variable constituted of 6 sub-factors by adding them. And you can't present Cronbach's alpha for variables for which unidimentionality has not been verified (i.e. job stress, burnout, resilience).
Moreover, some sub-factors present very low Cronbach's alpha (physical environment (0.313), lack of job autonomy (0.205). Those can't be used as sub-factors for further analysis.
Author Response
Response to Reviewer1.
Thank you for your kind recommendation.
Thank you for revisions and including Cronbach's apha data for sub-factors (note that vocabulary should be unified, as a "sub-factor" and an "item" could not be used indiferently - a sub-factor is constituted of items - ).
- But even if Cronbach's alpla are presented for sub-factors, this does not allow to analyse each variable as a whole. Ex: for the Scale of Burnout, there are 3 sub-factors (emotional exhaustion, cynicism and job achievement reduction). Those are constructs of items that should be analysis separately. You can not added them in a single construct named as Burnout, like it as been done in Tables 2 and 3. It is the same of KOSS and K-CD-RISC. For KOSS, initial study (Jang, 2005) created a tool of 43 items from which they extracted 8 sub-factors. You used 6 of them (30 items) for the present study. You have to analyse the results in light of 6 variables, and not of only 1 variable constituted of 6 sub-factors by adding them. And you can't present Cronbach's alpha for variables for which unidimentionality has not been verified (i.e. job stress, burnout, resilience).
: I referred to the thesis. (Jang, 2005)
We deleted 2 items from the Organized System, 1 item from the Insufficient Job Control, 6(total) items from the Job Security, and 4(total) items from the Occupational climate.
Because it was not suitable for the nursing system, so we deleted it.
Also, 123 lines, 136 lines, and 144 lines were deleted according to the feedback.
The references are as follows.
- Moreover, some sub-factors present very low Cronbach's alpha (physical environment (0.313), lack of job autonomy (0.205). Those can't be used as sub-factors for further analysis.
: Under COVID-19 specific situation, we included physical environment (0.313), lack of job autonomy (0.205), even if Cronbach's alpha were very low.
Thanks very much.

Reviewer 3 Report
I believe that the author has made sufficient revisions to improve the quality of the paper.
Author Response
Thanks very much.